# Genetic architecture of heart mitochondrial proteome influencing cardiac hypertrophy

**Karthickeyan Chella Krishnan[1]\*, Elie-Julien El Hachem[2], Mark P Keller[3], Sanjeet G Patel[4], Luke Carroll[5], Alexis Diaz Vegas[5], Isabela Gerdes Gyuricza[6], Christine Light[7], Yang Cao[8], Calvin Pan[8], Karolina Elżbieta Kaczor-Urbanowicz[9,10], Varun Shravah[11], Diana Anum[12], Matteo Pellegrini[10], Chi Fung Lee[7,13], Marcus M Seldin[14,15], Nadia A Rosenthal[6], Gary A Churchill[6], Alan D Attie[3], Benjamin Parker[16], David E James[5], Aldons J Lusis[8,17,18]\***

[1]Department of Pharmacology and Systems Physiology, University of Cincinnati College of Medicine, Cincinnati, United States; [2]Department of Integrative Biology and Physiology, Field Systems Biology, Sciences Sorbonne Université, Paris, France; [3]Biochemistry Department, University of Wisconsin-Madison, Madison, United States; [4]Department of Surgery/Division of Cardiac Surgery, University of Southern California Keck School of Medicine, Los Angeles, United States; [5]Metabolic Systems Biology Laboratory, Charles Perkins Centre, School of Life and Environmental Sciences, University of Sydney, Sydney, Australia; [6]Jackson Laboratory, Bar Harbor, United States; [7]Cardiovascular Biology Research Program, Oklahoma Medical Research Foundation, Oklahoma City, United States; [8]Department of Medicine/Division of Cardiology, University of California, Los Angeles, Los Angeles, United States; [9]Division of Oral Biology and Medicine, UCLA School of Dentistry, Los Angeles, United States; [10]UCLA Institute for Quantitative and Computational Biosciences, Los Angeles, United States; [11]Department of Chemistry, University of California, Los Angeles, United States; [12]Department of Integrative Biology and Physiology, University of California, Los Angeles, United States; [13]Department of Physiology, University of Oklahoma Health Sciences Center, Oklahoma City, United States; [14]Center for Epigenetics and Metabolism, Irvine, United States; [15]Department of Biological Chemistry, University of California, Irvine, United States; [16]Department of Anatomy and Physiology, University of Melbourne, Melbourne, Australia; [17]Department of Human Genetics, University of California, Los Angeles, United States; [18]Department of Microbiology, Immunology and Molecular Genetics, University of California, Los Angeles, United States

**\*For correspondence:** chellakn@ucmail.uc.edu (KCK); JLusis@mednet.ucla.edu (AJL)

**Abstract** Mitochondria play an important role in both normal heart function and disease etiology. We report analysis of common genetic variations contributing to mitochondrial and heart functions using an integrative proteomics approach in a panel of inbred mouse strains called the Hybrid Mouse Diversity Panel (HMDP). We performed a whole heart proteome study in the HMDP (72 strains, n=2-3 mice) and retrieved 848 mitochondrial proteins (quantified in ≥50 strains). High-resolution association mapping on their relative abundance levels revealed three *trans*-acting genetic loci on chromosomes (chr) 7, 13 and 17 that regulate distinct classes of mitochondrial proteins as well as cardiac hypertrophy. DAVID enrichment analyses of genes regulated by each of the loci revealed that the chr13 locus was highly enriched for complex-I proteins (24 proteins,

$P$=2.2E-61), the chr17 locus for mitochondrial ribonucleoprotein complex (17 proteins, $P$=3.1E-25) and the chr7 locus for ubiquinone biosynthesis (3 proteins, $P$=6.9E-05). Follow-up high resolution regional mapping identified NDUFS4, LRPPRC and COQ7 as the candidate genes for chr13, chr17 and chr7 loci, respectively, and both experimental and statistical analyses supported their causal roles. Furthermore, a large cohort of Diversity Outbred mice was used to corroborate *Lrpprc* gene as a driver of mitochondrial DNA (mtDNA)-encoded gene regulation, and to show that the chr17 locus is specific to heart. Variations in all three loci were associated with heart mass in at least one of two independent heart stress models, namely, isoproterenol-induced heart failure and diet-induced obesity. These findings suggest that common variations in certain mitochondrial proteins can act in *trans* to influence tissue-specific mitochondrial functions and contribute to heart hypertrophy, elucidating mechanisms that may underlie genetic susceptibility to heart failure in human populations.

## Editor's evaluation

This paper describing the genetic architecture of the heart mitochondrial proteome is important in that it identifies the major role of mitochondria in this process. The data is convincing, and appropriate and validated methodology in line with current state-of-the-art is used. This paper will be of interest to several groups of audiences, including cardiovascular researchers and geneticists.

## Introduction

Mitochondrial functions play a major role in the pathophysiology of several metabolic syndrome traits including obesity, insulin resistance and fatty liver disease (*Yin et al., 2014*; *Chouchani et al., 2014*; *Begriche et al., 2006*; *Sanyal et al., 2001*; *Kim et al., 2008*). There is now substantial evidence showing that genetic variation in mitochondrial functions contribute importantly to 'complex' diseases such as cardiovascular diseases (CVD) (*Wallace, 2018*). A scientific statement from the American Heart Association summarizes the central role of mitochondria in heart disease (*Murphy et al., 2016*). Mitochondrial bioenergetics and other functions impinge on virtually all aspects of the cell, including energy, cellular redox, apoptosis, and substrates for epigenetic modifications. Mitochondrial dysfunction also contributes to the development of heart failure (*Brown et al., 2017*; *Zhou and Tian, 2018*). The primary putative mechanism linking mitochondrial dysfunction to heart failure is decreased oxidative respiration leading to contractile failure. However, many other mechanisms have been postulated in recent years to implicate mitochondrial dysfunction in heart failure. They include excessive oxidative stress leading to inflammation, cell damage, and cell death; disturbed calcium homeostasis that triggers the opening of the mitochondrial permeability transition pore (mPTP), leading to loss of membrane potential, and eventual cell death. Therefore, mitochondria are an attractive target for heart failure therapy (*Brown et al., 2017*). Despite evidence showing that genetic variation in mitochondrial proteins is linked to disease, most studies tend to overlook the role of genetic variation in exploring the link between mitochondrial function and relevant phenotypes. To this end, we employ a 'systems genetics' approach to address this issue.

Our laboratory uses a combination of genetics, molecular biology, and informatics to investigate pathways underlying common cardiovascular and metabolic disorders. We exploit natural genetic variation among inbred strains of mice (and among human populations where possible) to identify novel targets and formulate hypotheses, taking advantage of unbiased global multi-omics technologies, such as transcriptomics, metabolomics, and proteomics, to help decipher causal mechanisms that drive complex traits. This 'systems genetics' approach integrates natural genetic variation with omics-level data (such as global protein abundance levels) to examine complex interactions that are difficult to address directly in humans. It shares with systems biology a holistic, global perspective (*Civelek and Lusis, 2014*; *Seldin et al., 2019*). Typically, a genetic reference population is examined for relevant clinical traits as well as global molecular traits, such as proteomics, and the data are integrated using correlation structure, genetic mapping, and mathematical modeling (*Civelek and Lusis, 2014*; *Seldin et al., 2019*).

Our systems genetics approach utilizes a well-characterized genetic reference population of 100 inbred mice strains, termed the Hybrid Mouse Diversity Panel (HMDP) (*Lusis et al., 2016*). The design of the HMDP resource, consisting of a panel of permanent inbred strains of mice that can be examined

for many phenotypes, has proved invaluable for studying metabolic syndrome traits. Major advantages of this approach are that the mapping resolution for complex traits is superior to traditional genetic crosses and the use of inbred strains affords replication of biological measures. It also facilitates studies of gene-by-environment and gene-by-sex interactions, which are difficult to address in human populations. Using this resource, we have characterized several cardiometabolic traits in both sexes over the last 10 years (*Parks et al., 2013*; *Parks et al., 2015*; *Rau et al., 2015*; *Org et al., 2016*; *Wang et al., 2016*; *Seldin et al., 2017*; *Rau et al., 2017*; *Norheim et al., 2017*; *Norheim et al., 2018*; *Hui et al., 2018*; *Seldin et al., 2018*; *Chella Krishnan et al., 2018*; *Chella Krishnan et al., 2019*; *Norheim et al., 2019*; *Chella Krishnan et al., 2021*).

In the present study, we have explored the genetic regulation of mitochondrial pathways and their contribution to heart function in the HMDP. Using an integrative proteomics approach, we now report the identification of three independent genetic loci that control distinct classes of mitochondrial proteins as well as heart hypertrophy. Each locus contains proteins previously shown to affect heart pathophysiology but by unknown mechanisms. Our results show that genetic diversity in the mitochondrial proteome plays a central role in heart pathophysiology.

## Results
### Genetic architecture of the heart mitochondrial proteome revealed three *trans*-regulatory hotspots

To investigate the effects of genetics on the heart proteome, we first performed a whole heart proteomic analysis in the HMDP (72 strains, n=2–3 mice, female sex; listed in *Supplementary file 1a*) and surveyed mitochondrial localization using MitoCarta3.0 (*Rath et al., 2021*). We retrieved abundance data for 848 of these proteins (quantified in ≥50 strains) and performed high-resolution association mapping on their respective abundance levels to the HMDP genotypes. Genetic variants associated to the protein abundance of nearby genes (<1 Mb) were referred to as *cis*- protein quantitative trait loci (pQTLs) and the remainder were referred to as *trans*-pQTLs. When a *trans*-pQTL locus is associated to multiple proteins, we defined it as a *trans*-pQTL hotspot. Using these criteria, we identified three hotspots, located on chromosome (chr) 7, chr13 and chr17, respectively (*Figure 1A*).

To identify the aspects of mitochondrial metabolism regulated by these loci, we constructed co-expression protein networks using weighted gene co-expression network analysis (WGCNA; *Langfelder and Horvath, 2008*) and identified five modules (*Figure 1B* and *Supplementary file 1b*). Eigengenes, representing the first principal component of two of these modules (Brown and Green), mapped to the same regions as the chr13 and chr17 loci, respectively. DAVID enrichment analyses (*Huang et al., 2009*) revealed that the chr13 locus (26 proteins; *Figure 2A* and *Supplementary file 1c*) that overlapped with Brown module (67 proteins, 96% overlap) was highly enriched for mitochondrial

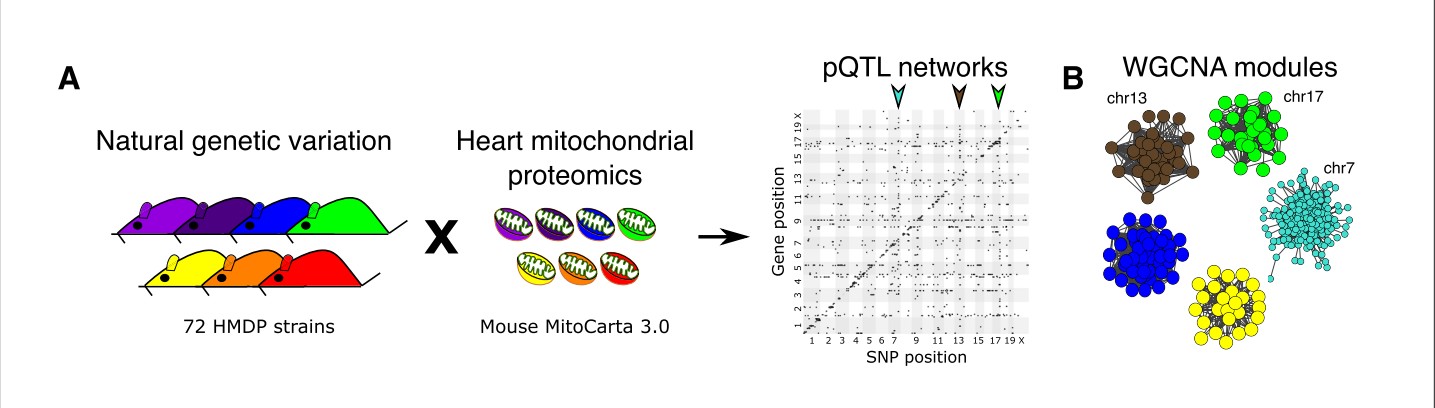

**Figure 1.** Genetic architecture of heart mitochondrial proteome. (**A**) High-resolution association mapping of 848 heart mitochondrial proteins from 72 HMDP strains to identify pQTL networks. Associations between protein abundance levels and genetic variants located within 1 Mb of the respective gene location were considered as *cis*-pQTLs (p<1E-05) shown along the diagonal axis and the rest were considered as *trans*-pQTLs (p<1E-06). Three *trans*-pQTL hotspots are indicated by arrows. (**B**) Five WGCNA modules and the respective *trans*-pQTL hotspots are shown. HMDP, hybrid mouse diversity panel; pQTL, protein quantitative trait locus; WGCNA, weighted gene co-expression network analysis.

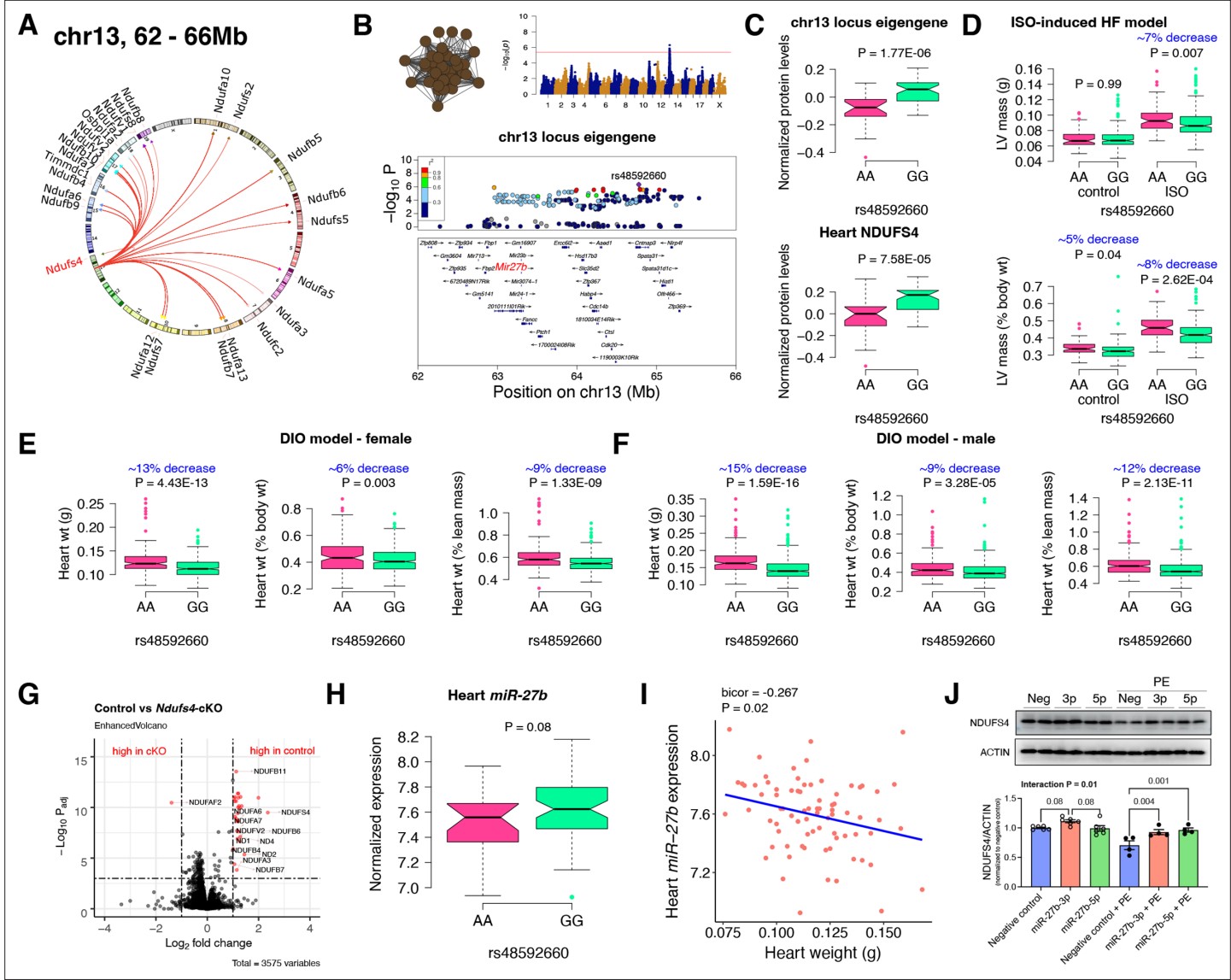

**Figure 2.** Chr13 locus controls mitochondrial complex-I *via miR-27b*/NDUFS4 axis. (**A**) Circos plot showing chr13 hotspot. Each line signifies a significant association between the genetic variants and the respective protein levels with candidate genes being highlighted. (**B**) Manhattan and regional plots of chr13 locus (brown module) eigengene, respectively. Red line signifies genome-wide significance threshold (p<4.1E-06). The peak SNPs and the candidate genes are highlighted. Genotype distribution plots of (**C**) protein levels and eigengenes, and cardiac phenotypes from (**D**) ISO-induced HF model and (**E**) female and (**F**) male DIO model at peak SNP (rs48592660) associated with chr13 locus. (**G**) Volcano plot showing differentially expressed protein levels between control and heart-specific *Ndufs4*-cKO mice (n=5 mice/group). Significantly different proteins (significance cutoff: abs[log$_2$FC]>1 and P$_{adj}$ <0.001) are highlighted in red. (**H**) Genotype distribution plots of heart *miR-27b* expression at peak SNP (rs48592660) associated with chr13 locus. (**I**) Gene-by-trait correlation plot between heart weight phenotype and heart *miR-27b* expression. (**J**) Immunoblot analyses of NDUFS4 protein levels in NRVMs transfected with mature *miR-27b* in the presence or absence of PE treatment. Data are presented as (**C–F and H**) boxplots showing median and interquartile range with outliers shown as circles (n=68–72 strains for protein levels; n=92–95 strains for ISO-model; n=92–100 strains for DIO-model; n=78 strains for miRNA levels) or (**J**) mean ± SEM (n=4–6 per group). p Values were calculated using (**A and B**) FaST-LMM that uses Likelihood-Ratio test; (**C–F and H**) Unpaired two-tailed Student's t test; (**I**) BicorAndPvalue function of the WGCNA R-package that uses Unpaired two-tailed Student's t test; (**J**) two-factor ANOVA corrected by post-hoc "Holm-Sidak's" multiple comparisons test. ISO, isoproterenol; HF, heart failure; DIO, diet-induced obesity; NRVMs, neonatal rat ventricular myocytes; PE, phenylephrine.

The online version of this article includes the following source data for figure 2:

**Source data 1.** Uncropped blots for *Figure 2*, panel J.

**Source data 2.** Raw unedited and uncropped blots for *Figure 2*, panel J.

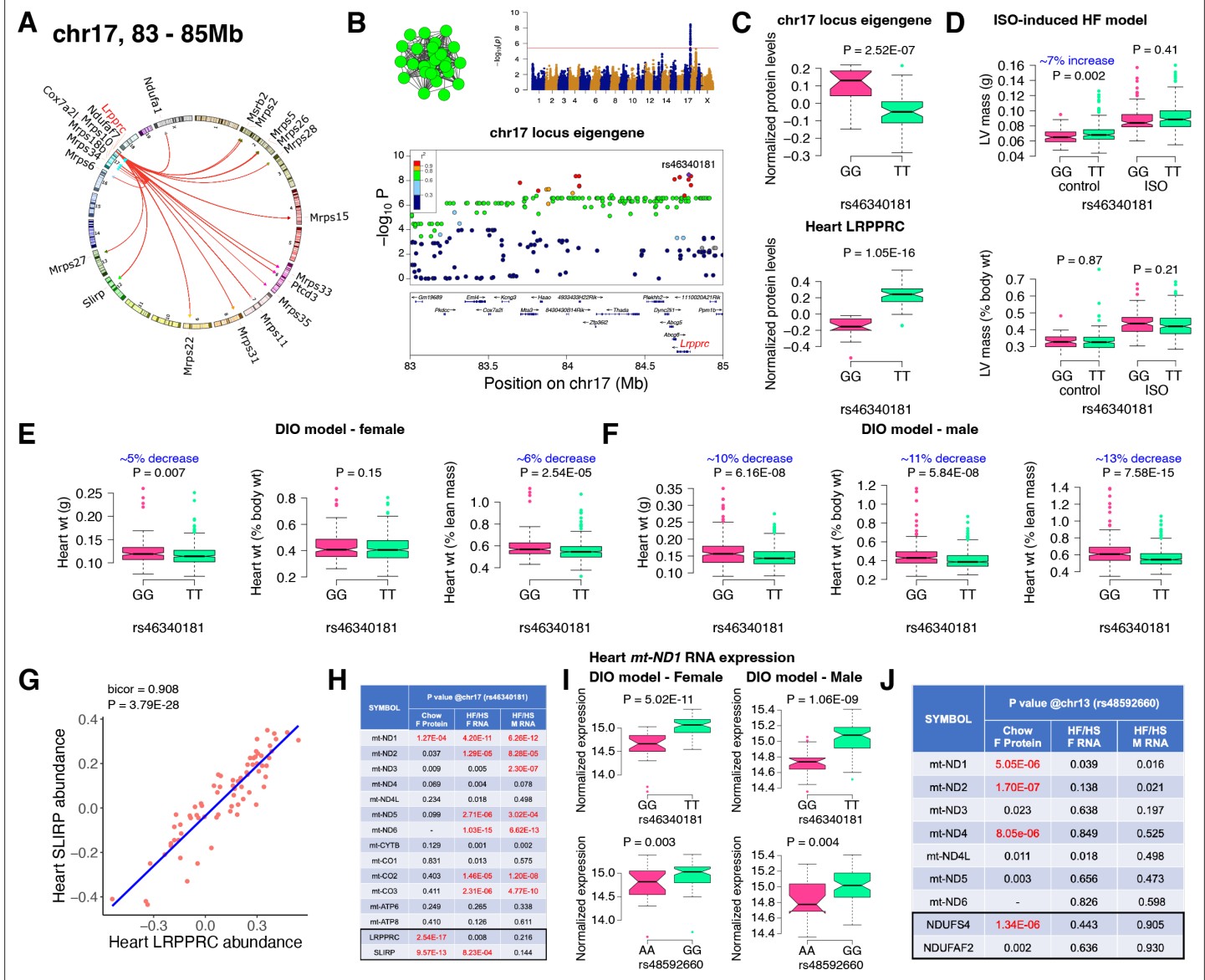

**Figure 3.** Chr17 locus controls mitoribosomes *via* LRPPRC/SLIRP. (**A**) Circos plot showing chr17 hotspot. Each line signifies a significant association between the genetic variants and the respective protein levels with candidate genes being highlighted. (**B**) Manhattan and regional plots of chr17 locus (green module) eigengene, respectively. Red line signifies genome-wide significance threshold (*P*<4.1E-06). The peak SNPs and the candidate genes are highlighted. Genotype distribution plots of (**C**) protein levels and eigengenes, and cardiac phenotypes from (**D**) ISO-induced HF model and (**E**) female and (**F**) male DIO model at peak SNP (rs46340181) associated with chr17 locus. (**G**) Protein-by-protein correlation plot between heart LRPPRC and SLIRP abundance levels. (**H**) Association p values between mtDNA-encoded mRNA expression or protein abundance levels and peak SNP (rs46340181) associated with chr17 locus in both sexes of HMDP. (**I**) Genotype distribution plots of heart *mt-ND1* mRNA expression from female and male DIO model at peak SNPs associated with chr17 (rs46340181) or chr13 (rs48592660) loci, respectively. (**J**) Association p values between mtDNA-encoded complex-I related mRNA expression or protein abundance levels and peak SNP (rs48592660) associated with chr13 locus in both sexes of HMDP. Data are presented as (**C–F and I**) boxplots showing median and interquartile range with outliers shown as circles (n=68–72 strains for protein levels; n=92–95 strains for ISO-model; n=92–100 strains for DIO-model). p Values were calculated using (**A, B, H and J**) FaST-LMM that uses Likelihood-Ratio test; (**C–F and I**) Unpaired two-tailed Student's t test; (**G**) BicorAndPvalue function of the WGCNA R-package that uses Unpaired two-tailed Student's t test.

The online version of this article includes the following source data for figure 3:

**Source data 1.** Raw data for *Figure 3*, panel H.

**Source data 2.** Raw data for *Figure 3*, panel J.

complex-I proteins (24 proteins, p=2.2E-61), and the chr17 locus (22 proteins; *Figure 3A* and *Supplementary file 1d*) that overlapped with Green module (38 proteins, 73% overlap) was highly enriched for mitochondrial ribonucleoprotein complex proteins (17 proteins, p=3.1E-25). The hotspot proteins in the chr7 locus (26 proteins; *Figure 4A* and *Supplementary file 1e*) were found primarily in the Turquoise module (402 proteins, 81% overlap) and was highly enriched for ubiquinone biosynthesis (3 proteins, p=6.9E-05).

## Chr13 locus controls mitochondrial complex-I

First, we analyzed the chr13 locus (26 proteins) that was highly enriched for mitochondrial complex-I (*Figure 2A*). Mapping the eigengene of the chr13 *trans*-regulated proteins identified the peak SNP (rs48592660). NDUFS4, a protein critical for complex-I assembly and loss of which leads to cardiac hypertrophy (*Chouchani et al., 2014*) mapped near the locus but outside the region of linkage disequilibrium. However, within the locus was the *microRNA(miR)–23b/27b/24–1* cluster, among which *miR-27b*, a conserved regulator of NDUFS4, was identified *via* nine miRNA target prediction algorithms and one dataset of experimentally validated miRNA targets (*Figure 2B* and *Supplementary file 1f*). Cardiac overexpression of *miR-27b* has previously been shown to promote cardiac hypertrophy (*Wang et al., 2012*) but attenuate angiotensin II-induced atrial fibrosis (*Wang et al., 2018*). We therefore hypothesized that the chr13 locus regulated complex-I proteins by influencing *miR-27b* and thus NDUFS4 protein levels. Indeed, we observed higher levels of both the chr13 locus eigengene and NDUFS4 with the GG allele of the peak locus SNP (*Figure 2C*). To identify the functional relevance of this peak SNP, we analyzed its association in two independent heart stress models, namely, isoproterenol (ISO)-induced heart failure (*Wang et al., 2016*) and diet-induced obesity (DIO) (*Parks et al., 2013*) models. We observed significantly lower left ventricular mass under ISO stress (*Figure 2D*) and lower heart weight under DIO stress in both sexes (*Figure 2E*) of strains harboring the GG allele, thus confirming the directionality of genetic impacts on NDUFS4 protein levels and hypertrophic response.

## NDUFS4 heart-specific knockout mice had reduced mitochondrial complex-I proteins

NDUFS4 is an 18 kDa accessory subunit that is essential for the mitochondrial complex-I assembly (*Zhu et al., 2016*; *Stroud et al., 2016*; *Gu et al., 2016*; *Kahlhöfer et al., 2017*; *Scacco et al., 2003*). Loss-of-function mutations in NDUFS4 leads to complex-I deficiency causing a neuromuscular disease, Leigh syndrome (*Leshinsky-Silver et al., 2009*) and is also involved in cardiomyopathies (*Chouchani et al., 2014*; *Karamanlidis et al., 2013*; *Zhang et al., 2018*). Taken together, we wanted to experimentally test our hypothesis that NDUFS4 protein independently controls the complex-I protein abundance levels in heart. For this, we performed whole heart proteomic analyses in both control and heart-specific *Ndufs4*-cKO mice (n=5 mice/group). Among the abundance data for 3575 proteins, we observed 31 proteins to be significantly different between the control and cKO groups (significance cutoff: abs[log$_2$FC]>1 and P$_{adj}$ <0.001; listed in *Supplementary file 1g*). Strikingly, 21 of these proteins were found in the chr13 locus and these were highly enriched for complex-I proteins (30 proteins, p=3.8E-80). Notably, only one protein, NDUFAF2, was up regulated in *Ndufs4*-cKO mice (*Figure 2G*). This is intriguing because NDUFAF2 has been reported to stabilize complex-I in the absence of NDUFS4 and to sustain complex-I activity (*Adjobo-Hermans et al., 2020*; *Leong et al., 2012*) indicating this may be a compensatory mechanism.

## *MiR-27b* controls NDUFS4 protein levels and heart weights

As an independent corroboration and to understand the consequence of the chr13 locus on *miR-27b* expression, we independently sequenced miRNAs from DIO-stressed female HMDP strains (n=78 strains). We found that strains harboring the GG allele had higher *miR-27b* expression (*Figure 2H*), and there was a significant inverse correlation between heart weights and *miR-27b* expression in these mice (*Figure 2I*). Based on these observations, we hypothesize that *miR-27b* increases NDUFS4 protein levels thereby reducing heart weights. To validate this observation, we transfected neonatal rat ventricular myocytes (NRVMs) with mature *miR-27b* in the presence or absence of phenylephrine (PE) treatment. The precursor miR-27b has two mature arms: miR-27b-5p and miR-27b-3p that originates from the 5' or 3' strand of the precursor miR-27b, respectively and are almost complementary to

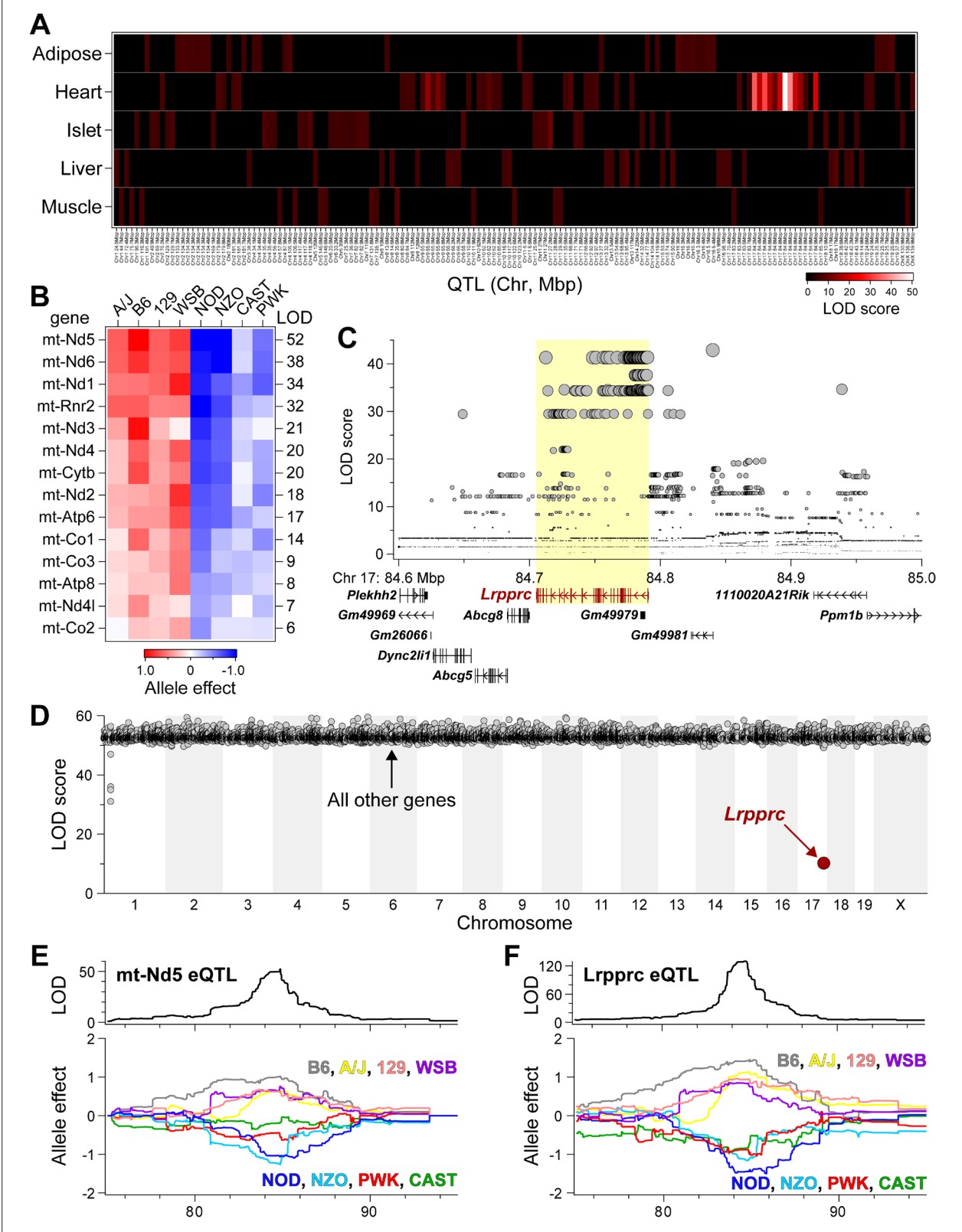

**Figure 4.** *Lrpprc* regulates an mt-encoded eQTL hotspot specifically in heart of DO mice. (**A**) Heatmap illustrates all eQTL (LOD >6) for mt-encoded transcripts that were identified in Adipose (***Wang et al., 2012***), Heart (***Ruzzenente et al., 2012***), Islet (***Gu et al., 2016***), Liver (***Rath et al., 2021***), and Muscle (***Wang et al., 2016***) from Diversity Outbred mice maintained on a Western-style diet. Mt-eQTL are arranged along x-axis according to genomic position from chr1 to chrX; z-axis depicts LOD score, highlighting a hotspot on chr17 at ~85 Mbp. (**B**) Allele effect patterns for mt-eQTL mapping to

*Figure 4 continued on next page*

*Figure 4 continued*

chr17 hotspot in heart. Red illustrates alleles associated with increased expression; blue, decreased expression. LOD scores are shown along right margin. (**C**) SNP association profile for mt-Nd5 eQTL in heart. *Lrpprc* contains SNPs with strongest association, yellow highlighted region. (**D**) Mediation of mt-Nd5 eQTL against all transcripts in heart. The LOD score for mt-Nd5 is significantly reduced when conditioned on a *Lrpprc* cis-eQTL, consistent with genetic regulation of *Lrpprc* being required for the regulation of mt-Nd5. The mt-Nd5 eQTL (**E**) and *Lrpprc* cis-eQTL (**F**) in heart demonstrate matched and concordant allele effect patterns, suggesting *Lrpprc* is a positive driver of mt-Nd5.

The online version of this article includes the following figure supplement(s) for figure 4:

**Figure supplement 1.** Mediation analysis identifies Lrpprc as a driver for the heart-specific mt- eQTL hotspot on chr17.

each other. The stability and functionality of each arm depends on the tissue/cell type and according to miRBase Sequence database (*Kozomara and Griffiths-Jones, 2011*) (database of published miRNA sequences and annotation), miR-27b-3p (98.8%) is more abundant than miR-27b-5p (1.2%) and therefore is assumed to be the stable and functional version. Immunoblot analyses revealed that NDUFS4 protein levels were reduced with PE treatment in the control cells but *miR-27b-3p* consistently increased NDUFS4 protein levels in both control and PE-treated conditions while *miR-27b-5p* increased NDUFS4 protein levels only in the PE-treated conditions (*Figure 2J*). Taken together, we conclude that the chr13 locus affects mitochondrial complex-I proteins through the *miR-27b*/NDUFS4 axis, thereby controlling heart weights.

## Chr17 locus controls mitoribosomes

Next, we analyzed the chr17 locus (22 proteins) that was highly enriched for mitochondrial ribosomal proteins (*Figure 3A*). We mapped the eigengene of the significantly associated mitochondrial proteins to identify the peak SNP (rs46340181). We identified LRPPRC as a candidate as it was the only protein controlled in cis by the peak SNP (*Figure 3B*). Importantly, LRPPRC together with SLIRP controls mitochondrial mRNA stability, enabling polyadenylation and translation (*Siira et al., 2017*; *Chujo et al., 2012*; *Ruzzenente et al., 2012*). Loss-of-function mutations in LRPPRC cause a congenital mitochondrial disease called Leigh syndrome, French-Canadian type that is often characterized by mitochondrial complex IV deficiency and impaired mitochondrial respiration and in some cases, neonatal cardiomyopathy and congenital cardiac abnormalities have been reported (*Mootha et al., 2003*; *Oláhová et al., 2015*). It is noteworthy that SLIRP is also under the control of the chr17 locus (*Figure 3A*). Further, our phenotypic associations revealed that the eigengene and LRPPRC were inversely associated with the TT allele (*Figure 3C*). This was functionally translated into lower heart weight in strains harboring the TT allele in both sexes under DIO stress only (*Figure 3D–F*). We also observed that abundance levels of both LRPPRC and SLIRP proteins were strongly correlated with each other (*Figure 3G*) and controlled by the chr17 locus (*Figure 3A*), thus demonstrating that they are co-regulated.

## LRPPRC/SLIRP protein complex controls mitochondrial transcript levels

Based on our current observations and published data, we hypothesized that high LRPPRC/SLIRP protein complex stabilizes mitochondrial transcripts, thus reducing the need to upregulate mitochondrial translation. To test this, we independently sequenced the heart transcripts from our HMDP mice that underwent DIO stress. We observed that the chr17 locus peak SNP (rs46340181), which strongly controls both LRPPRC and SLIRP proteins, is only associated with the mitochondrial mRNA expression and not their respective protein levels (*Figure 3H*). Moreover, this phenomenon was observed in both sexes, explaining the lack of sex bias in phenotypic associations (*Figure 3E and F*). In contrast, the chr13 locus that controlled complex-I proteins did not show strong associations with transcript levels in either sex (*Figure 3I and J*). As a specific example, *Figure 3I* shows that the mRNA expression of *mt-ND1* was higher in strains harboring the TT allele in both sexes under DIO stress, illustrating that upregulated LRPPRC/SLIRP is stabilizing the mitochondrial transcript (*Figure 3C*), resulting in reduced heart weights (*Figure 3D–F*).

## Corroboration with Diversity Outbred mice also identifies a heart-specific chr17 mt-eQTL hotspot controlled by LRPPRC

As independent corroboration, we utilized the Diversity Outbred (DO) mice (*Svenson et al., 2012*) to further explore our chr17 locus. To extend our observations about the genetics of mitochondrially encoded (mt-encoded) gene regulation, we asked if mt-encoded genes are subjected to genetic regulation in multiple tissues from a large cohort of DO mice. We used RNA-sequencing to survey gene expression that included transcripts encoded by the mitochondrial genome, in five tissues (adipose, heart, islet, liver and skeletal muscle) from ~500 DO mice that were maintained on a Western-style diet high in fat and sucrose. All DO mice were genotyped at >140 K SNPs with the GigaMUGA microarray (*Morgan et al., 2015*), enabling eQTL analysis of the mt-encoded transcripts. Among the five tissues surveyed, we identified 157 mt-eQTL (LOD >6) for 15 mt-encoded genes, including all 13 protein-coding genes, and the two genes that encode for ribosomal RNA proteins (*Supplementary file 1h*). Given that the mitochondrial and nuclear genomes are distinct, mt-encoded eQTL at nuclear loci reflect trans-acting mechanisms that bridge the two genomes (*Ali et al., 2019*).

While all tissues yielded multiple mt-eQTL, heart demonstrated a striking hotspot where 14 mt-encoded genes uniquely mapped to chr17 at ~85 Mbp with LOD scores ranging from 52 (mt-Nd5) to 6 (mt-Co2) (*Figure 4A*; listed in *Supplementary file 1h*). The founder allele-effect signatures for the 14 mt-eQTL at the hotspot were the same and partitioned the eight haplotypes into 'high' (A/J, B6, 129, WSB) versus 'low' (NOD, NZO, CAST, PWK) subgroups (*Figure 4B*), suggesting a common causal variant behind the co-mapping of the mt-eQTL. The SNP association profile for the mt-Nd5 eQTL identified a block of SNPs directly over the *Lrpprc* gene locus (*Figure 4C*), including one missense variant (rs33393440, Lys466Glu), which is present in the high allele subgroup (*Supplementary file 1i*). Taken together, these results suggest that SNPs within *Lrpprc* may be responsible for the heart-specific mt-eQTL hotspot.

We used mediation analysis to identify potential causal gene underlying physiological or molecular QTL as described (*Keller et al., 2016*; *Keller et al., 2018*; *Keller et al., 2019*). In mediation analysis of gene expression, trans-eQTL are conditioned on the expression of all other genes, including those at the locus to which the trans-eQTL map. If the genetic signal of the trans-eQTL decreases upon conditioning of a specific gene, that gene becomes a strong candidate as the driver of the trans-eQTL. We focused on the trans-eQTL for mt-Nd5 at the chr17 locus in heart, as this demonstrated the strongest genetic signal (*Figure 4B*). Mediation against *Lrpprc* resulted in a large drop in the LOD score for mt-Nd5 eQTL (*Figure 4D*). Similar results were observed for mediation of the other mt-eQTL to the heart hotspot (*Figure 4—figure supplement 1*). The allele effect patterns of the mt-Nd5 eQTL (*Figure 4E*) and that for the *Lrpprc* eQTL (*Figure 4F*) demonstrate the same, and concordant high and low haplotype subgroups. Taken together, our results in DO mice are consistent with that in the HMDP mouse cohort, and strongly suggest that LRPPRC functions as a positive driver of the mt-eQTL hotspot in heart.

## Chr7 locus affects CoQ metabolism

Finally, we analyzed the chr7 locus (26 proteins), which unlike the chr13 or chr17 loci, had no major representation of a single mitochondrial protein complex but was moderately enriched for ubiquinone biosynthesis (*Figure 5A*). Mapping the eigengene of the significantly associated mitochondrial proteins identified the peak SNP (rs32451909). We identified COQ7 as a strong candidate as it was the only protein exhibiting a *cis*-regulation at the locus in the region of linkage disequilibrium (*Figure 5B*). COQ7 catalyzes a critical step in the biosynthesis of coenzyme Q (CoQ). Among several functions, CoQ participates in electron transport facilitating ATP synthesis. CoQ also has a clear role in heart failure (*Sharma et al., 2016*). Phenotypically, we observed lower levels of both the eigengene and COQ7 with the GG allele (*Figure 5C*). This was functionally translated into higher heart weight in strains containing GG allele in both sexes under DIO stress only (*Figure 5D–F*). We also observed that abundance levels of other COQ proteins were strongly correlated with COQ7 protein, thus demonstrating that they are co-regulated (*Figure 5G–I*). At least two of these proteins, COQ3 and COQ6, are associated with the chr7 locus (*Figure 5A*). When we measured the CoQ levels in both the mitochondrial fractions and total heart lysates from DIO-stressed female HMDP strains (n=15 strains), we observed a significant upregulation in the levels of both CoQ9 and CoQ10 in the lysates from HMDP

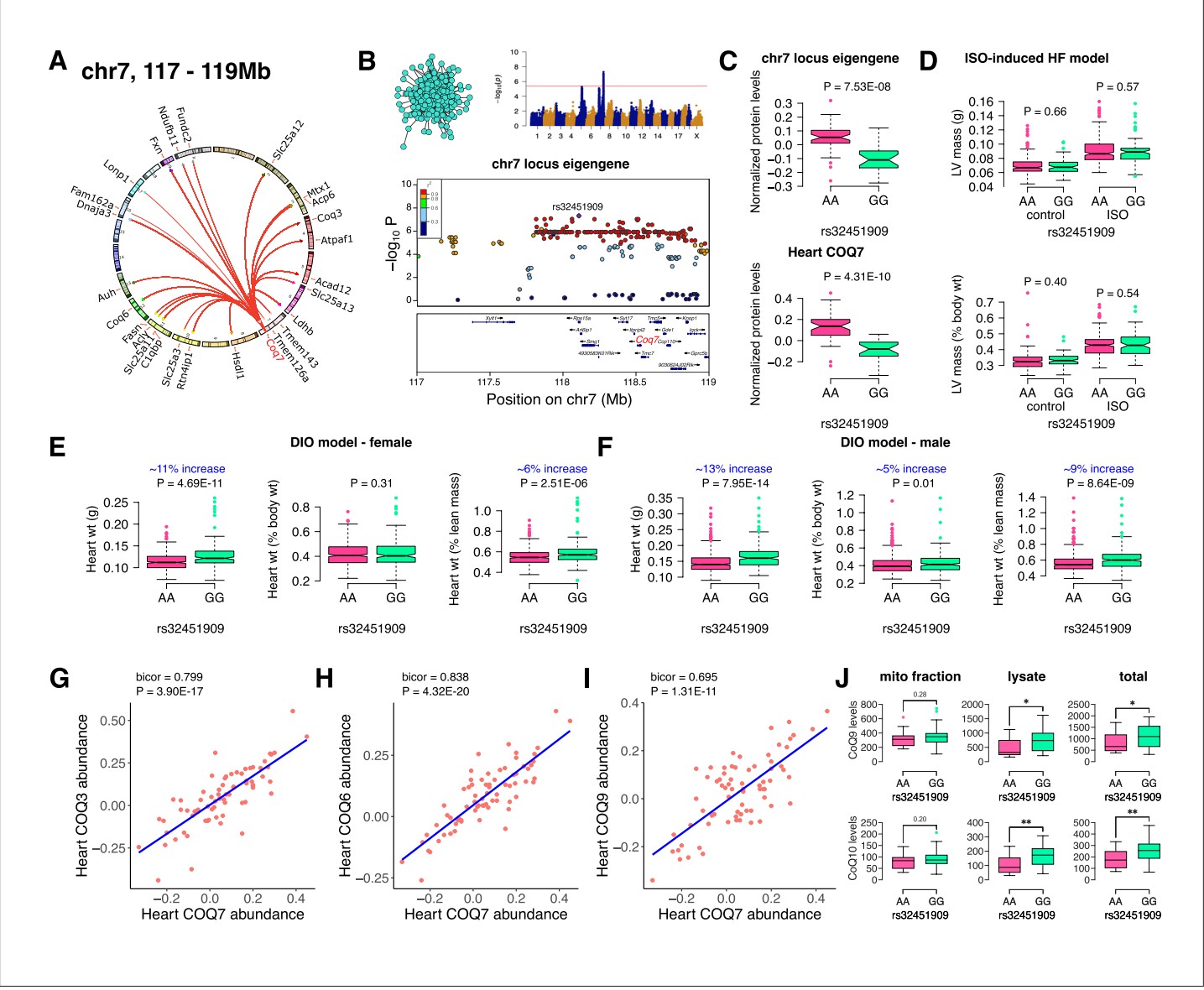

**Figure 5.** Chr7 locus affects CoQ metabolism *via* COQ7. (**A**) Circos plot showing chr7 hotspot. Each line signifies a significant association between the genetic variants and the respective protein levels with candidate genes being highlighted. (**B**) Manhattan and regional plots of chr7 locus (turquoise module) eigengene, respectively. Red line signifies genome-wide significance threshold (p<4.1E-06). The peak SNPs and the candidate genes are highlighted. Genotype distribution plots of (**C**) protein levels and eigengenes, and cardiac phenotypes from (**D**) ISO-induced HF model and (**E**) female and (**F**) male DIO model at peak SNP (rs32451909) associated with chr7 locus. Protein-by-protein correlation plots between heart COQ7 and (**G**) COQ3, (**H**) COQ6 and (**I**) COQ9 abundance levels. (**J**) Genotype distribution plots of CoQ9 or CoQ10 levels in both the mitochondrial fractions and total heart lysates from female DIO model at peak SNPs associated with chr7 (rs32451909) locus. Data are presented as (**C–F and J**) boxplots showing median and interquartile range with outliers shown as circles (n=68–72 strains for protein levels; n=92–95 strains for ISO-model; n=92–100 strains for DIO-model; n=15 strains for CoQ9 and CoQ10 levels). p Values were calculated using (**A and B**) FaST-LMM that uses Likelihood-Ratio test; (**C–F and J**) Unpaired two-tailed Student's t test; (**G–I**) BicorAndPvalue function of the WGCNA R-package that uses Unpaired two-tailed Student's t test.

mice harboring the GG allele (*Figure 5J*). Taken together, we conclude that the chr7 locus controls CoQ metabolism *via* regulation of the COQ7 protein.

## Discussion

We have previously used systems genetics analyses in HMDP and discovered a central role for adipose mitochondrial function in the sex differences observed in cardiometabolic traits, including

obesity, insulin resistance and plasma lipids (*Norheim et al., 2019*; *Chella Krishnan et al., 2021*), and liver mitochondrial function in non-alcoholic fatty liver disease (*Chella Krishnan et al., 2018*). In the present study, an integrative proteomics approach was utilized in HMDP to investigate the effects of genetic regulation of mitochondrial pathways on heart function. Several conclusions have emerged. First, we identified three distinct and independent *trans*-regulating genetic loci, on chr13, chr17 and chr7, respectively. Second, our enrichment analyses suggested mechanisms perturbed by each of these loci. Thus, the chr13 locus was enriched for mitochondrial complex I, chr17 for mitoribosomes, and chr7 was enriched for ubiquinone biosynthesis. Third, our regional mapping identified NDUFS4, LRPPRC and COQ7 as candidate genes for chr13, chr17, and chr7 loci, respectively. Finally, we performed both experimental and statistical analyses to support their causal roles. Each of these points is discussed in detail below.

*Trans*-acting human pQTLs, mostly using blood/plasma proteome (*Yao et al., 2018*; *Ruffieux et al., 2020*; *Zhong et al., 2020*; *Yang et al., 2021*; *Folkersen et al., 2020*; *He et al., 2020*), and *trans*-acting human eQTLs (*Aguet, 2017*; *Brynedal et al., 2017*; *Yao et al., 2017*; *Small et al., 2018*; *GTEx Consortium, 2020*), have been reported, but exploring *trans*-regulatory landscapes in human studies is limited by sample size, tissue accessibility, and environmental factors. In the current study, mapping genetic loci controlling protein abundance levels in heart tissues from an inbred mouse population revealed that ~9% of available nuclear-encoded mitochondrial proteins (74/848) are significantly controlled (p<1E-6) by three genetic loci, on chr13 (26 proteins), chr17 (22 proteins) and chr7 (26 proteins), respectively. Each of these three *trans* regulating hotspots were specific for the proteome, as they were not identified in our heart transcriptome. Interestingly, two of these hotspots independently controlled mitochondrial complex I, one at the mRNA (chr17 *via* LRPPRC/SLIRP) and the other at the protein level (chr13 *via* miR-27b/NDUFS4). Finally, the chr7 locus was found to control ubiquinone metabolism *via* COQ7 protein.

Mitochondrial complex I is the first and largest (~1 MDa) protein complex of the mitochondrial electron transport chain comprised of 45 subunits, seven encoded in the mtDNA (*Zhu et al., 2016*; *Fiedorczuk et al., 2016*; *Hirst, 2013*). NDUFS4 is an 18 kDa accessory subunit that assembles the catalytic N-module (where NADH oxidation occurs) with the rest of the complex I. Mutations leading to the loss of NDUFS4 result in complex I deficiency (*Zhu et al., 2016*; *Stroud et al., 2016*; *Gu et al., 2016*; *Kahlhöfer et al., 2017*; *Scacco et al., 2003*) and are involved in cardiomyopathies, both in protective (ischemia/reperfusion injury) and detrimental roles (pressure overload) (*Chouchani et al., 2014*; *Karamanlidis et al., 2013*; *Zhang et al., 2018*). It was also reported that loss of NDUFS4 causes reduced complex I mediated ROS generation (ischemia/reperfusion injury) and increased protein acetylation (pressure overload), leading to their respective pathologies. Here, we demonstrate that heart-specific NDUFS4 KO mice have higher levels of NDUFAF2, which is reported to partly stabilize complex I in the absence of NDUFS4, although the resultant complex I is unstable and has reduced activity (*Adjobo-Hermans et al., 2020*). Taken together, we report that natural variation in NDUFS4 protein levels affect other complex I proteins resulting in reduced complex I activity and cardiac hypertrophy. On the other hand, the role of *miR-27b* on heart function is unclear. Cardiac-specific overexpression of *miR-27b* has been reported to increase pressure overload-induced cardiac hypertrophy *via* PPARG (*Wang et al., 2012*) but attenuated angiotensin II-induced atrial fibrosis *via* ALK5 (*Wang et al., 2018*). Also, whole body knockout of *miR-27b* was found to attenuate pressure overload-induced cardiac hypertrophy *via* FGF1 (*Li et al., 2022*), but this could be due to extracardiac effects mediated by *miR-27b-3p* such as adipocyte browning (*Yu et al., 2018*; *Sun and Trajkovski, 2014*; *Kong et al., 2015*). Using 100 HMDP inbred strains of mice, here we found that cardiac *miR-27b* expression was inversely correlated with heart weights and that mature *miR-27b-3p* appears to have a protective role in reducing cardiac hypertrophy *via* increasing NDUFS4 protein levels.

Mitochondria are unique organelles having their own genome that encodes for 13 protein coding genes, which are essential subunits of complexes I, III, IV, and V. Nuclear-encoded LRPPRC protein working together with the SLIRP protein helps to enable mitochondrial mRNA translation *via* promoting its polyadenylation and stability (*Siira et al., 2017*; *Chujo et al., 2012*; *Ruzzenente et al., 2012*). Loss-of-function mutations in LRPPRC causes Leigh syndrome, French-Canadian type that are associated with complex IV deficiency, impaired mitochondrial respiration and in some cases, neonatal cardiomyopathy, and congenital cardiac abnormalities (*Mootha et al., 2003*; *Oláhová et al., 2015*). Here, we show that natural genetic variations in a panel of inbred mouse strains at a chr17 locus (TT

allele) result in higher protein levels of the LRPPRC/SLIRP complex as well as lower heart weights. This phenomenon is most likely mediated by increased expression levels of mitochondrial mRNA that coincide with reduced protein levels of mitoribosomes. We propose that since these mitochondrial mRNAs are protected by LRPPRC/SLIRP complex, fewer mitoribosomes are necessary for translation. Interestingly, although LRPPRC mutations are often associated with complex IV deficiency (*Mootha et al., 2003*; *Oláhová et al., 2015*), we observed that complex I mRNAs were also strongly regulated by the LRPPRC/SLIRP complex, in addition to complex IV mRNAs. Thus, further research on the bioenergetic consequences of LRPPRC/SLIRP loss-of-function mutations beyond complex IV deficiency are warranted.

Mitochondrial coenzyme Q, or ubiquinone or CoQ, is essential for mitochondrial electron transport chain function as it shuttles electrons from both complexes I and II to complex III. The enzyme COQ7 is responsible for catalyzing the penultimate step in CoQ biosynthesis. Here, we report that the chr7 locus (GG allele) is associated with reduced levels of COQ7 protein, which is in turn associated with increased heart weights. Interestingly, when we measured CoQ levels in these hearts, we found no differences in the mitochondrial fractions but an increase in both CoQ9 and CoQ10 levels in the lysates. Mice with a complete loss of *Coq7* expression have embryonic lethality (*Nakai et al., 2001*) but survive for several months after a knockout is induced in the adult (*Wang et al., 2015*). Interestingly, *Coq7* heterozygous knockout mice are long-lived but exhibit dysfunctional mitochondria (such as reduced respiration, reduced ATP levels and increased ROS generation) with no differences in CoQ levels despite significantly reduced COQ7 protein levels (*Lapointe and Hekimi, 2008*; *Lapointe et al., 2009*; *Liu et al., 2005*). Furthermore, these *Coq7* heterozygous knockout mice were found to harbor a varying distribution of CoQ9 levels in their liver mitochondria with the outer membrane having increased CoQ9 while the inner membrane had reduced CoQ9 levels (*Lapointe et al., 2012*). It was postulated that higher CoQ9 levels in the liver mitochondrial outer membrane may be a protective response to increased ROS levels generated in *Coq7* heterozygous knockout mice (*Lapointe et al., 2012*). Thus, reduced COQ7 protein levels do not necessarily associate with reduced CoQ levels but have heterogeneous sub-mitochondrial and possibly subcellular CoQ distributions. Taken together, we propose that increased heart weights in mice harboring chr7 locus (GG allele) is a result of increased ROS generation caused by reduced COQ7 protein levels. Also, the increased CoQ levels in the heart lysates of these mice might be a protective response against the oxidative stress generated by reduced COQ7 protein in these mice. Further research is warranted in understanding the purported role of COQ7 in submitochondrial and subcellular CoQ heterogeneous distribution focusing on heart pathophysiology.

In conclusion, our unbiased systems genetics analyses identified three loci regulating mitochondrial function in the heart. None of these loci were observed when transcript levels were examined, providing justification for proteomic rather than transcriptomic studies. All three loci are associated with heart mass in two independent heart stress models and/or negatively correlated with known hypertrophic markers (*Supplementary file 1j*). Nevertheless, the percent change in heart mass between the three loci were only of modest value. However, both heart failure and hypertrophy being complex traits are influenced by a large number of genetic factors, each exerting a small to large effect. The resulting phenotypes are therefore a sum of all impacts. So, we believe it is important to characterize modest as well as large effects. In summary, our results provide mechanistic information about the roles of previously studied genes namely, NDUFS4, LRPPRC, and COQ7, in heart failure.

## Methods
### Mice
All mice were purchased from The Jackson Laboratory and bred at UCLA according to approved institutional animal care and use committee (IACUC) protocols with daily monitoring. Both the ISO-induced HF (*Rau et al., 2015*; *Wang et al., 2016*; *Rau et al., 2017*) and DIO (*Parks et al., 2013*; *Parks et al., 2015*) models were previously described in detail. Briefly, for ISO-induced HF model, 8–10 weeks of age female mice were administered with isoproterenol (30 mg per kg body weight per day, Sigma) for 21 days using ALZET osmotic minipumps, which were surgically implanted intraperitoneally. For DIO model, mice were fed ad libitum a chow diet (Ralston Purina Company) until 8 weeks of age and then placed ad libitum on a high-fat/high-sucrose (HF/HS) diet (Research Diets-D12266B, New Brunswick,

NJ) with 16.8% kcal protein, 51.4% kcal carbohydrate, 31.8% kcal fat for an additional 8 weeks. For heart proteomic analysis, 8–12 weeks of age female HMDP mice (72 strains, n=2–3 mice; listed in *Supplementary file 1a*) and 3–5 months old control and *Ndufs4*-cKO mice of both sex (n=5 each group) fed with chow diet ad libitum (Purina 5053, LabDiet) were used. All mice were maintained on a 14 hr light/10 hr dark cycle (light is on between 6 a.m. and 8 p.m.) at a temperature of 25 degrees and 30–70% humidity. On the day of the experiment, the mice were sacrificed after 4 hr fasting.

## Studies in Diversity Outbred mice

All experiments that used Diversity Outbred (DO) mice were conducted in the Biochemistry Department at the University of Wisconsin-Madison and approved by the university's Animal Care and Use Committee as described previously (*Keller et al., 2018*). Briefly, 4-week-old DO mice were purchased from the Jackson Labs (stock no. 009376) and placed on a Western-style diet high in fat and sucrose (17.3% protein, 34% carbohydrate, and 44.6% kcal fat) from Envigo Teklad (TD.08811). Mice were sacrificed at ~20 weeks of age, and tissues harvested and prepared for RNA-sequencing and expression quantitative trait loci (eQTL) analysis as previously detailed (*Keller et al., 2018*). The expression of the 13 protein-coding mt-genes (mt-Nd1, mt-Nd2, mt-Nd3, mt-Nd4, mt-Nd4l, mt-Nd5, mt-Nd6, mt-Cytb, mt-Co1, mt-Co2, mt-Co3, and mt-Atp6, mt-Atp8) and two mt-ribosomal RNAs (mt-Rnr1, mt-Rnr2) were detected in each of the five tissues surveyed (heart, islet, skeletal muscle, adipose, and liver). Expression values for these mt-encoded transcripts in all tissues, and *Lrpprc* in heart, are provided in *Supplementary file 2*.

## Heart global proteomic analysis

Heart tissue from the HMDP, control and *Ndufs4*-cKO mice were lysed in 6 M guanidine HCL (Sigma; #G4505), 100 mM Tris pH 8.5 containing 10 mM tris(2-carboxyethyl)phosphine (Sigma; #75259) and 40 mM 2-chloroacetamide (Sigma; #22790) by tip-probe sonication. The lysate was heated at 95 °C for 5 min and centrifuged at 20,000 x *g* for 10 min at 4 °C. The supernatant was diluted 1:1 with water and precipitated overnight with five volumes of acetone at –20 °C. The lysate was centrifuged at 4000 x *g* for 5 min at 4 °C and the protein pellet was washed with 80% acetone. The lysate was centrifuged at 4000 x *g* for 5 min at 4 °C and the protein pellet was resuspended in Digestion Buffer (10% 2,2,2-Trifluoroethanol [Sigma; #96924] in 100 mM HEPEs pH 8.5). Protein was quantified with BCA (ThermoFisher Scientific) and normalized in Digestion Buffer to a final concentration of 2 µg/µl. Protein was digested with sequencing grade trypsin (Sigma; #T6567) and sequencing grade LysC (Wako; #129–02541) at a 1:50 enzyme:substrate ratio overnight at 37 °C with shaking at 2000 x rpm. Eight micrograms of peptide was directly labelled with 32 µg of 10-plex TMT (lot #QB211242) in 20 µl at a final concentration of 50% acetonitrile for 1.5 hr at room temperature. The reaction was de-acylated with a final concentration of 0.3% (w/v) hydroxylamine and quenched with a final concentration of 1% trifluoroacetic acid (TFA). Each 10-plex experiment contained nine different strains with a tenth reference label (131 isobaric label) made up of the same peptide digest from pooled mix of C57BL/6 J heart. Following labelling, the peptides from each of the 18 TMT 10-plex batches were pooled and purified directly by Styrene Divinylbenzene - Reversed-Phase Sulfonate (SDB-RPS) microcolumns, washed with 99% isopropanol containing 1% TFA and eluted with 80% acetonitrile containing 2% ammonium hydroxide followed by vacuum concentration. Peptides were resuspended in 2% acetonitrile containing 0.1% TFA and thirty micrograms of peptide was fractionated on an in-house fabricated 25 cm x 320 µm column packed with C18BEH particles (3 µm, Waters). Peptides were separated on a gradient of 0–30% acetonitrile containing 10 mM ammonium formate (pH 7.9) over 60 min at 6 µl/min using an Agilent 1260 HPLC and detection at 210 nm with a total of 48 fractions collected and concatenated down to 12 fractions.

## Mass spectrometry and data processing

Peptide fractions from heart were resuspended in 2% acetonitrile containing 0.1% TFA and analyzed on a Dionex ultra-high pressure liquid chromatography system coupled to an Orbitrap Lumos mass spectrometer. Briefly, peptides were separated on 40 cm x 75 µm column containing 1.9 µm C18AQ Reprosil particles on a linear gradient of 2–30% acetonitrile over 2 h. Electrospray ionization was performed at 2.3 kV with 40% RF lens and positively charged peptides detected via a full scan MS (350–1550 m/z, 1e6 AGC, 60 K resolution, 50ms injection time) followed data-dependent MS/MS

analysis performed with CID of 35% normalized collision energy (NCE) (rapid scan rate, 2e4 AGC, 50ms injection time, 10ms activation time, 0.7 m/z isolation) of the top 10 most abundant peptides. Synchronous-precursor selection with MS3 (SPS-MS3) analysis was enabled with HCD of 60 NCE (100–500 m/z, 50 K resolution, 1e5 AGC, 105ms injection time) (*McAlister et al., 2014*). Dynamic exclusion was enabled for 60 s. Data were processed with Proteome Discoverer v2.3 and searched against the Mouse UniProt database (November 2018) using SEQUEST (*Eng et al., 1994*). The precursor MS tolerance were set to 20 ppm and the MS/MS tolerance was set to 0.8 Da with a maximum of 2 miss-cleavage. The peptides were searched with oxidation of methionine set as variable modification, and TMT tags on peptide N-terminus / lysine and carbamidomethylation of cysteine set as a fixed modification. All data was searched as a single batch and the peptide spectral matches (PSMs) of each database search filtered to 1% FDR using a target/decoy approach with Percolator (*Käll et al., 2007*). The filtered PSMs from each database search were grouped and q-values generated at the peptide level with the Qvality algorithm (*Käll et al., 2009*). Finally, the grouped peptide data was further filtered to 1% protein FDR using Protein Validator. Quantification was performed with the reporter ion quantification node for TMT quantification based on MS3 scans in Proteome Discoverer. TMT precision was set to 20 ppm and corrected for isotopic impurities. Only spectra with <50% co-isolation interference were used for quantification with an average signal-to-noise filter of >10. The data was filtered to retain Master proteins that were measured in at least 50 mice.

## HMDP heart mRNA and miRNA expression analysis

Using the miRNeasy Mini Kit (QIAGEN), total RNA was extracted from HMDP heart tissues. From this, global mRNA expression were analyzed as previously described (*Cao et al., 2022*), while the QIAseq miRNA Library Kit (QIAGEN) was used to create miRNA libraries. These libraries were then sequenced using 1x50 HiSeq sequencing. Reads were mapped using hisat2 (version 2.0.6) and counted using htseq-count (version 0.13.5). Differential expression analysis was performed with DESeq2.

## Association mapping

Genotypes for the mouse strains were obtained using the Mouse Diversity Array (*Bennett et al., 2010*). After filtering for quality or missing genotypes, about 200,000 remained. Genome-wide association for phenotypes and protein abundance levels was performed using Factored Spectrally Transformed Linear Mixed Models (FaST-LMM), which applies a linear mixed model to correct for population structure (*Lippert et al., 2011*). A cutoff value for genome-wide significance was set at 4.1E-06, as determined previously (*Bennett et al., 2010*).

## Cell culture and treatments

Following isolation, neonatal rat ventricular myocytes (NRVMs) were plated in DMEM containing 10% Fetal bovine serum (FBS) and 1% antibiotics overnight. The next day, NRVMs were changed to serum-free medium in the presence or absence of 100 µM phenylephrine (Sigma, Cat# P6126-10G). The cells were then transfected with miRNA control, mature *miR-27b-5p*, or *miR-27b-3p* mimics (Sigma) for 48 hr. The cell lysates were harvested for immunoblotting using primary antibodies against NDUFS4 (# sc-100567, Santa Cruz) and Actin (# 8457 S, Cell Signaling). Band densitometry was quantified using ImageJ Gel Plugin (NIH).

| Reagent | Mature sequence 5′ to 3′ |
| --- | --- |
| MISSION microRNA, Negative Control 1 (miRNA control) | GGUUCGUACGUACACUGUUCA |
| MISSION microRNA - hsa-miR-27b* (miR-27b-5p) | AGAGCUUAGCUGAUUGGUGAAC |
| MISSION microRNA - hsa-miR-27b (miR-27b-3p) | UUCACAGUGGCUAAGUUCUGC |

## Mitochondria isolation from frozen hearts

Twenty mg of frozen hearts were thawed in 1.4 mL of ice-cold isolation buffer (70 mM sucrose, 220 mM mannitol, 1 mM EGTA, 2 mM HEPES, pH 7.4 containing protease inhibitors). Hearts were mechanically homogenized with 20 strokes of a Dounce homogenizer at 4 °C. Homogenates were centrifuged at

1000 $g$ for 10 min at 4°C. Supernatant was collected (300 µL were reserved as whole lysate) and centrifuged at 10,000 $g$ for 10 min at 4°C to obtain a pellet containing the mitochondria. Mitochondrial pellet was re-suspended in 1 mL of isolation buffer and re-centrifuged at 10,000 $g$ for 10 min at 4°C. The mitochondrial pellet was finally re-suspended in 200 µL of isolation buffer and protein concentration determined using the BCA assay.

## CoQ extraction from whole lysate and mitochondrial enriched fractions

CoQ extraction, detection and analysis was performed as described previously (*Burger et al., 2020*). Briefly, 15 µg protein from mitochondrial fractions or 200 µg protein from total lysate were aliquoted into 2 mL Eppendorf tubes and volumes adjusted to 100 µL for mitochondrial fractions or 200 µL for whole lysates, and samples were kept on ice for the entire process. 20 µL of 0.1 ng/mL CoQ8 (2 ng total; Avanti Polar Lipids) were added to each sample as an internal standard. To protein aliquots ice-cold 250 µL acidified methanol (0.1% HCl in MeOH) were added to each sample, followed by 300 µL hexane and samples were thoroughly mixed by vertexing. Hexane and MeOH/water layers were separated by centrifugation at 15,000 $g$ for 5 min at 4°C. The upper layer of hexane was collected and transferred to clean 2 mL Eppendorf tubes and completely dried in a GeneVac vacuum centrifuge, on a low BP point method for 40 min (20 min pre-final stage and 20 min final stage). Samples were reconstituted in 100 µL ethanol before analysis of CoQ levels by LC-MS.

## LC-MS analysis of CoQ

CoQ8, 9 and 10 levels were analyzed using a TSQ Altis triple quadrupole mass spectrometer (ThermoFisher) coupled to a Vanquish LC system (ThermoFisher). Fifteen µL of sample was injected and separated on a 2.6 µm Kinetex XB-C18 100 A column (50×2.10 mm; Phenomenex) at 45°C. Mobile Phase A consisted of 2.5 mM ammonium formate in 95% MeOH, 5% IPA and Mobile Phase B consisted of 2.4 mm ammonium formate in 100% IPA. A gradient method over 5 min was used with an initial concentration of 0% B held for 1 min before increased to 45% B over 1 min and held for 1 min, before decreasing back to 0% B over 0.5 min and column re-equilibrated over 1.5 min. Eluent was then directed into the QQQ with the following settings: source voltage = 3500 V; sheath gas 2; aux gas 2; transfer capillary temperature = 350 °C. Ammonium adducts of each of the analytes were detected by SRM with Q1 and Q3 resolution set to 0.7 FWHM with the following parameters: [CoQ8 +NH4]$^+$, $m/z$ 744.9 → 197.1 with collision energy 32.76; [CoQ9 +NH4]$^+$, $m/z$ 812.9 → 197.1 with collision energy 32.76; [CoQ9H$_2$+NH4]$^+$, $m/z$ 814.9 → 197.1 with collision energy 36.4; [CoQ10 +NH4]$^+$, $m/z$ 880.9 → 197.1 with collision energy 32.76; and [CoQ10H$_2$+NH4]$^+$, $m/z$ 882.9 → 197.1 with collision energy 36.4. CoQ9 and CoQ10 were quantified in samples against a standard curve (0–1,000 nM) and normalized to spiked in CoQ8 levels (20 ng/mL) in each sample and standard. No CoQ9H$_2$ and CoQ10H$_2$ were detected in any of the samples, though some bleed through into these channels was detected for the oxidized analytes, though these are resolved from the reduced standard peaks (as reported previously).

## Statistical analysis

Statistical analyses were performed using Prism v9.4.0 (GraphPad Software, Inc, La Jolla, CA, USA). Errors bars plotted on graphs are presented as the mean ± SEM unless reported otherwise. The critical significance value (α) was set at 0.05, and if the p values were less than α, we reported that the observed differences were statistically significant. Correlations were calculated using biweight midcorrelation using the bicorAndPvalue function of the WGCNA package (*Langfelder and Horvath, 2008*).

## Acknowledgements

This work was supported by NIH grants DK120342, HL148577 and HL147883 (AJL) and DOD grant W81XWH2110115 (AJL); R00DK120875 (KCK); R00HL138193 (MMS); R01DK101573, R01DK102948, and RC2DK125961 (ADA); Wisconsin Alumni Research Foundation (MPK); National Health and Medical Research Council of Australia (NHMRC) grants and fellowships (DEJ and BLP); Systems Biology Association fellowship, Foundation Sorbonne fellowship, French Minister, and Master BIP (EJEH). The funders had no role in study design, data collection and interpretation, or the decision to submit the work for publication.

# Additional information

## Competing interests

David E James: Senior Editor, eLife; Senior Editor for the Systems Genetics Special Issue (DEJ had no involvement in the decision making for this submission). The other authors declare that no competing interests exist.

## Funding

| Funder | Grant reference number | Author |
| --- | --- | --- |
| National Institutes of Health | DK120342 | Aldons J Lusis |
| National Institutes of Health | R00DK120875 | Karthickeyan Chella Krishnan |
| National Institutes of Health | R00HL138193 | Marcus M Seldin |
| National Institutes of Health | R01DK101573 | Alan D Attie |
| U.S. Department of Defense | W81XWH2110115 | Aldons J Lusis |
| Wisconsin Alumni Research Foundation | | Mark P Keller |
| National Health and Medical Research Council | | Benjamin Parker |
| National Institutes of Health | R01DK102948 | Alan D Attie |
| National Institutes of Health | RC2DK125961 | Alan D Attie |
| National Institutes of Health | HL148577 | Aldons J Lusis |
| National Institutes of Health | HL147883 | Aldons J Lusis |
| Systems Biology Association fellowship | | Elie-Julien El Hachem |
| Foundation Sorbonne fellowship | | Elie-Julien El Hachem |
| French Minister and Master BIP | | Elie-Julien El Hachem |

The funders had no role in study design, data collection and interpretation, or the decision to submit the work for publication.

## Author contributions

Karthickeyan Chella Krishnan, Conceptualization, Data curation, Formal analysis, Supervision, Funding acquisition, Validation, Investigation, Visualization, Methodology, Writing - original draft, Project administration, Writing – review and editing; Elie-Julien El Hachem, Yang Cao, Calvin Pan, Marcus M Seldin, Formal analysis, Investigation, Visualization; Mark P Keller, Luke Carroll, Alexis Diaz Vegas, Karolina Elżbieta Kaczor-Urbanowicz, Formal analysis, Investigation, Visualization, Methodology; Sanjeet G Patel, Formal analysis, Investigation, Methodology; Isabela Gerdes Gyuricza, Christine Light, Varun Shravah, Diana Anum, Formal analysis, Investigation; Matteo Pellegrini, Chi Fung Lee, Nadia A Rosenthal, Gary A Churchill, Alan D Attie, Resources, Supervision, Funding acquisition; Benjamin Parker, Formal analysis, Investigation, Visualization, Methodology, Writing – review and editing; David E James, Resources, Supervision, Funding acquisition, Writing – review and editing; Aldons J Lusis, Conceptualization, Resources, Supervision, Funding acquisition, Project administration, Writing – review and editing

### Author ORCIDs
Karthickeyan Chella Krishnan http://orcid.org/0000-0002-4895-2495
Elie-Julien El Hachem http://orcid.org/0000-0001-5033-6117
Marcus M Seldin http://orcid.org/0000-0001-8026-4759
Gary A Churchill http://orcid.org/0000-0001-9190-9284
David E James http://orcid.org/0000-0001-5946-5257

### Ethics
Mice.All mice were purchased from The Jackson Laboratory and bred at UCLA according to approved institutional animal care and use committee (IACUC) protocols with daily monitoring. All mice were maintained on a 14 h light/10 h dark cycle (light is on between 6 a.m. and 8 p.m.) at a temperature of 25 degrees and 30-70% humidity. On the day of the experiment, the mice were sacrificed after 4-hour fasting.Studies in Diversity Outbred mice.All experiments that used Diversity Outbred (DO) mice were conducted in the Biochemistry Department at the University of Wisconsin-Madison and approved by the university's Animal Care and Use Committee. Briefly, 4-week-old DO mice were purchased from the Jackson Labs (stock no. 009376) and placed on a Western-style diet high in fat and sucrose (17.3% protein, 34% carbohydrate, and 44.6% kcal fat) from Envigo Teklad (TD.08811).

### Decision letter and Author response
Decision letter https://doi.org/10.7554/eLife.82619.sa1
Author response https://doi.org/10.7554/eLife.82619.sa2

## Additional files

### Supplementary files
• Supplementary file 1. Supplementary tables. (**a**) List of HMDP strains used for proteomic analyses and their respective genotypes at the three loci. (**b**) Five modules of WGCNA and their corresponding proteins. (**c**) List of proteins associated with chr13 locus listed in *Figure 2A*. The associated P values between the respective proteins and chr13 locus eigengene pSNP are listed. (**d**) List of proteins associated with chr17 locus listed in *Figure 3A*. The associated P values between the respective proteins and chr17 locus eigengene pSNP are listed. (**e**) List of proteins associated with chr7 locus listed in *Figure 4A*. The associated P values between the respective proteins and chr7 locus eigengene pSNP are listed. (**f**) Scoring of miRNA targeting *Ndufs4* by nine algorithms and one dataset of experimental validation. (**g**) List of differentially abundant proteins between the control and *Ndufs4*-cKO groups, and their P values shown in *Figure 2*. Only proteins that passed the signifcant cut-off (abs[log2FC]>1 and Padj <0.001) are listed. Proteins overlapping chr13 locus are shown in blue font. (**h**) List of 157 mt-eQTL (LOD >6) across five tissues in the DO mice as shown in *Figure 4A*. (**i**) SNP association profile for mt-Nd5 in heart as shown in *Figure 4C*. (**j**) Protein-protein correlations between the three candidates and known hypertrophic markers. The bicor P values between the respective markers and each locus candidate gene are listed.

• Supplementary file 2. Expression values for mt-encoded transcripts and/or *Lrpprc* in skeletal muscle (2 a), adipose (2b), islets (2 c), liver (2d) and heart (2e).

• Transparent reporting form

### Data availability
All sequencing raw data can be accessed at the Gene Expression Omnibus under accession GSE194198 (HF/HS heart HMDP RNA-seq data) and GSE207142 (HF/HS heart HMDP miRNA-seq data). The mass spectrometry proteomics data have been deposited to the ProteomeXchange Consortium via the PRIDE *Perez-Riverol et al., 2022* partner repository with the dataset identifiers PXD036398 (HMDP hearts) and PXD036399 (NDUFS4 KO hearts).

The following datasets were generated:

| Author(s) | Year | Dataset title | Dataset URL | Database and Identifier |
|---|---|---|---|---|
| Lusis AJ, Chella Krishnan K, Kaczor-Urbanowicz KE | 2022 | Small RNA sequencing of hearts from female mice on high fat diet, from 85 strains | https://www.ncbi.nlm.nih.gov/geo/query/acc.cgi?acc=GSE207142 | NCBI Gene Expression Omnibus, GSE207142 |
| Lusis AJ, Chella Krishnan K, Parker BL | 2022 | HMDP hearts | https://proteomecentral.proteomexchange.org/cgi/GetDataset?ID=PXD036398 | ProteomeXchange, PXD036398 |
| Lusis AJ, Chella Krishnan K, Parker BL | 2022 | NDUFS4 KO hearts | https://proteomecentral.proteomexchange.org/cgi/GetDataset?ID=PXD036399 | ProteomeXchange, PXD036399 |

The following previously published dataset was used:

| Author(s) | Year | Dataset title | Dataset URL | Database and Identifier |
|---|---|---|---|---|
| Lusis AJ, Cao Y | 2022 | Heart profiling in HMDP (high-fat/high-sucrose diet) | https://www.ncbi.nlm.nih.gov/geo/query/acc.cgi?acc=GSE194198 | NCBI Gene Expression Omnibus, GSE194198 |

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
