## [Editor Report]

This paper describing the genetic architecture of the heart mitochondrial proteome is important in that it identifies the major role of mitochondria in this process. The data is convincing, and appropriate and validated methodology in line with current state-of-the-art is used. This paper will be of interest to several groups of audiences, including cardiovascular researchers and geneticists.

---

## [Decision Letter]

**Decision letter after peer review:**

Thank you for submitting your article "Genetic Architecture of Heart Mitochondrial Proteome influencing Cardiac Hypertrophy" for consideration by *eLife*. Your article has been reviewed by 3 peer reviewers, one of whom is a member of our Board of Reviewing Editors, and the evaluation has been overseen by a Senior Editor. The following individual involved in the review of your submission has agreed to reveal their identity: Mathias Mericskay (Reviewer #3).

The reviewers have discussed their reviews with one another, and the Reviewing Editor has drafted this letter to help you prepare a revised submission.

Essential revisions:

• It is not clear, at least to this reviewer, why the authors focused on NDUFS4. In the Results section, the authors mention: "NDUFS4, a protein critical for complex-I assembly and loss of which leads to cardiac hypertrophy (2) mapped near the locus but outside the region of linkage disequilibrium. However, within the locus was the microRNA(miR)-23b/27b/24-1 cluster, among which miR-27b, a conserved regulator of NDUFS4, was identified via nine miRNA target prediction algorithms and one dataset of experimentally validated miRNA targets." If NDUFS4 is outside the range, why did the authors then focus on a microRNA that regulates this protein? The concept behind the focus on NDUFS4 is not clear and needs to be better explained.

• The western blot in Figure 2J is not convincing at all and the difference noted on the bar graph is not supported by the western blot.

• The authors do not describe the difference between mir27b-5p and mir27b-3p. This needs to be better described.

• Why does mir27b-5p decrease NDUFS4 compared to control in Figure 2J?

• It's not clear why the authors used MitoCarta 2.0 and not the more updated MitoCarta 3. An explanation for this decision would be appreciated. Regardless, it would be ideal to retroactively expand the analysis to MitoCarta 3.0.

Figure 2

• The correlation between miR-27b expression and heart weight is not impressive. Given that this is a key part of the mechanistic hypothesis underlying this correlative observation and the low magnitude and lack of statistical power of the effect in Figure 2H, this is a significant weakness.

• Apart from changes in heart size, it would be informative to assess hemodynamic cardiac characteristics such as ejection fraction to assess changes in cardiac function.

• Changes in protein abundance don't necessarily indicate changes in function, especially such small changes. Therefore, the addition of functional assays directly related to the protein would greatly enhance this paper. For example, the authors could examine changes in mitochondrial respiration or metabolism between treatments and genotypes.

• As a result of these issues, the authors' conclusion in lines 201-202 is not, in my opinion, adequately supported by the provided data. Figures 2H and 2I show a correlation. The cause-and-effect experiment of transfecting miR-27b into NRVM is not very impressive by itself. Adding a third replicate to the PE group may help show if there is a statistical difference; it's not clear why they have fewer replicates in this group. Primary ACM might be an alternative model system to show miR-27b increases NDUFS4 levels (although it would be more technically challenging). The researchers could also block miR-27b and if their hypothesis holds true should see a decrease in NDUF4.

Figure 3

• The lack of any functional assessments to strengthen the correlation, let alone actual causation experiments is clearly apparent. Please make the data for this figure to provide a functional assessment.

Overall

In lines 44-46 the impression was given that these three variations would cause heart weight changes in both ISO and DIO HF models. Unfortunately, only one of the three genes showed heart weight changes in both conditions.

Tissue slices showing the histology or functional imaging studies of the heart (both of which have been done previously by papers published by this group) would provide much stronger evidence that the genes they identify could play a role in causing/preventing heart failure. Are the changes in weight the authors see sufficient to cause/prevent heart failure? It would be a nice touch for the authors to include additional comments in this area in the discussion as some of the changes in weight are small.

• The size effect impact on cardiac hypertrophy is more difficult to appreciate as if I understand well only the data after ISO or DIO are shown for each allele but not baseline values. LV mass/body weight % of 0.4-0.45% as shown in Figure 2D does not suggest massive hypertrophy (0.4% = 100 mg LV for 25 g of body weight, for instance, can be seen in normal mice). The same holds true for figures 2E-F. I would like the authors to give the values of LV mass in untreated animals (is it similar between the 2 genotypes?) and maybe express the hypertrophy in relative increase to baseline in each group. In this way, one could better realize if the treatment was really efficient to trigger hypertrophy and how much the response differs between the groups.

• The same holds true for figures 3E-F and 4E-F: Is this clinically meaningful as the size effect seems to be very low? I would like the authors to discuss that point. Can the 3 loci have a cumulative effect? Is the study powered enough to detect that?

• Could the authors provide the level of other markers of hypertrophy from their protein screen such as BNP, Β-MHC (although difficult to distinguish from α-MHC by proteomic), or any other relevant protein? This would strengthen their hypothesis if the loci variant were also associated with quantitative differences in these classical markers of cardiac hypertrophy.

• sum up the values of mitochondria and lysate levels per individual to perform a statistical analysis of the total level of each coQ?

• If we admit that the total CoQ9 and CoQ10 levels are increased when COQ7 involved in their synthesis is reduced, it remains a puzzling observation. The authors discuss the possibility that CoQ is relocalized to the outer membrane or in the cytosol when COQ7 is reduced in heterozygotes CO7 mutant or in the mice bearing the rs32451909 GG allele, but this does not resolve the question of the total level of CoQ metabolite. Can the authors elaborate a bit further on that observation and by which mechanism it could occur? For instance, are the proteins involved in coQ degradation repressed when COQ7 is low?

*Reviewer #1 (Recommendations for the authors):*

This is a well-written paper and the data support the conclusion. As mentioned in the public review, the authors need to address a couple of concerns.

1. It is not clear, at least to this reviewer, why the authors focused on NDUFS4. In the Results section, the authors mention: "NDUFS4, a protein critical for complex-I assembly and loss of which leads to cardiac hypertrophy (2) mapped near the locus but outside the region of linkage disequilibrium. However, within the locus was the microRNA(miR)-23b/27b/24-1 cluster, among which miR-27b, a conserved regulator of NDUFS4, was identified via nine miRNA target prediction algorithms and one dataset of experimentally validated miRNA targets." If NDUFS4 is outside the range, why did the authors then focus on a microRNA that regulates this protein? The concept behind the focus on NDUFS4 is not clear and needs to be better explained.

2. The western blot in Figure 2J is not convincing at all and the difference noted on the bar graph is not supported by the western blot.

3. The authors do not describe the difference between mir27b-5p and mir27b-3p. This needs to be better described.

4. Why does mir27b-5p decrease NDUFS4 compared to control in Figure 2J?

*Reviewer #2 (Recommendations for the authors):*

Figure 1

– It's not clear why the authors used MitoCarta 2.0 and not the more updated MitoCarta 3. An explanation for this decision would be appreciated. Regardless, it would be ideal to retroactively expand the analysis to MitoCarta 3.0.

Figure 2

– The correlation between miR-27b expression and heart weight is not impressive. Given that this is a key part of the mechanistic hypothesis underlying this correlative observation and the low magnitude and lack of statistical power of the effect in Figure 2H, this is a significant weakness.

– Similar to the above, changes in NDUFS4 protein levels in NRVM cells in response to PE and miR-27b are not very convincing. The authors are careful in their interpretation of increased NDUFS4 in miR-27b-3p, not claiming significance, but do not mention miR-27b-5p. It would be nice to mention the results of miR-27b-5p in the text.

– Apart from changes in heart size, it would be informative to assess hemodynamic cardiac characteristics such as ejection fraction to assess changes in cardiac function.

– Repeating this experiment in a more physiologically relevant model like primary cultured adult cardiomyocytes may help improve these data.

– Changes in protein abundance don't necessarily indicate changes in function, especially such small changes. Therefore, the addition of functional assays directly related to the protein would greatly enhance this paper. For example, the authors could examine changes in mitochondrial respiration or metabolism between treatments and genotypes.

– As a result of these issues, the authors' conclusion in lines 201-202 is not, in my opinion, adequately supported by the provided data. Figures 2H and 2I show a correlation. The cause-and-effect experiment of transfecting miR-27b into NRVM is not very impressive by itself. Adding a third replicate to the PE group may help show if there is a statistical difference; it's not clear why they have fewer replicates in this group. Primary ACM might be an alternative model system to show miR-27b increases NDUFS4 levels (although it would be more technically challenging). The researchers could also block miR-27b and if their hypothesis holds true should see a decrease in NDUF4.

Figure 3

– The authors could comment on whether patients with Leigh syndrome, French-Canadian type suffer from cardiomyopathies.

– The lack of any functional assessments to strengthen the correlation, let alone actual causation experiments is clearly apparent.

Figure 4

– Again, the lack of any functional assessments to strengthen the correlation, let alone actual causation experiments is clearly apparent.

Overall

– Regarding hypertrophy, I was given unmet expectations about the impact. In lines 44-46 I got the impression that these three variations would cause heart weight changes in both ISO and DIO HF models. Unfortunately, only one of the three genes showed heart weight changes in both conditions.

– Regarding heart failure, at the end of the abstract and the beginning of the intro, the focus is on the potential impact of this paper for identifying genes that might underlie genetic susceptibility to heart failure. The authors are cautious with their claims, which is good, but it would be nice to see more than weight measurements and see metrics that indicate heart failure in addition to hypertrophy. Either tissue slices showing the histology or functional imaging studies of the heart (both of which have been done previously by papers published by this group) would provide much stronger evidence that the genes they identify could play a role in causing/preventing heart failure. I am left wondering if the weight differences are relevant to pathologic or physiologic adaptations. Are the changes in weight the authors see sufficient to cause/prevent heart failure? It would be a nice touch for the authors to include additional comments in this area in the discussion as some of the changes in weight are small.

– The authors did a good job with their conclusion wrapping up their results with previously published biology. They stated their limitations and it is clear that the genes they identified still require more validation. They also do a good job summarizing the significance of this paper and their research, which is identifying genes that were only possible because they were looking at proteomics within their HMDP.

*Reviewer #3 (Recommendations for the authors):*

1. I am not an expert in bioinformatics and pQTL identification, so I will not comment on that aspect and believe the hotspots identified are indeed strongly associated with the mitochondrial protein level. For instance, the size effect of the different rs48592660 allele on the protein level of the gene identified in the Ch13 locus is indeed important. However, the size effect impact on cardiac hypertrophy is more difficult to appreciate as if I understand well only the data after ISO or DIO are shown for each allele but not baseline values. LV mass/body weight % of 0.4-0.45% as shown in Figure 2D does not suggest massive hypertrophy (0.4% = 100 mg LV for 25 g of body weight, for instance, can be seen in normal mice). The same holds true for figure 2E-F. I would like the authors to give the values of LV mass in untreated animals (is it similar between the 2 genotypes?) and maybe express the hypertrophy in relative increase to baseline in each group. In this way, one could better realize if the treatment was really efficient to trigger hypertrophy and how much the response differs between the groups.

2. The same holds true for figures 3E-F and 4E-F, so finally, for all the loci described. My point is: Is this clinically meaningful as the size effect seems to be very low? I would like the authors to discuss that point. Can the 3 loci have a cumulative effect? Is the study powered enough to detect that?

3. Could the authors provide the level of other markers of hypertrophy from their protein screen such as BNP, Β-MHC (although difficult to distinguish from α-MHC by proteomic), or any other relevant protein? This would strengthen their hypothesis if the loci variant were also associated with quantitative differences in these classical markers of cardiac hypertrophy.

4. When looking at figure 4G, while CoQ9 and CoQ10 increase in the lysate, they do not decrease in the mitochondrial fraction, and if any trend should be seen, they rather increase there too, although I agree it's not significant. Could the authors sum up the values of mitochondria and lysate levels per individual to perform a statistical analysis of the total level of each coQ?

5. If we admit that the total CoQ9 and CoQ10 levels are increased when COQ7 involved in their synthesis is reduced, it remains a puzzling observation. The authors discuss the possibility that CoQ is relocalized to the outer membrane or in the cytosol when COQ7 is reduced in heterozygotes CO7 mutant or in the mice bearing the rs32451909 GG allele, but this does not resolve the question of the total level of CoQ metabolite. Can the authors elaborate a bit further on that observation and by which mechanism it could occur? For instance, are the proteins involved in coQ degradation repressed when COQ7 is low?

---

## [Author Response]

Essential revisions:• It is not clear, at least to this reviewer, why the authors focused on NDUFS4. In the Results section, the authors mention: "NDUFS4, a protein critical for complex-I assembly and loss of which leads to cardiac hypertrophy (2) mapped near the locus but outside the region of linkage disequilibrium. However, within the locus was the microRNA(miR)-23b/27b/24-1 cluster, among which miR-27b, a conserved regulator of NDUFS4, was identified via nine miRNA target prediction algorithms and one dataset of experimentally validated miRNA targets." If NDUFS4 is outside the range, why did the authors then focus on a microRNA that regulates this protein? The concept behind the focus on NDUFS4 is not clear and needs to be better explained.

The chr13 locus was highly enriched for mitochondrial complex-I, and among the three complex-I proteins encoded by chr13 (NDUFS4, NDUFS6 and NDUFAF2), only NDUFS4 protein levels were affected by chr13 locus. Therefore, we focused on NDUFS4.

• The western blot in Figure 2J is not convincing at all and the difference noted on the bar graph is not supported by the western blot.

We have independently repeated our experiments and have now revised the data with new blots and graphs (Figure 2J). We have also supplied source data of the blots as part of the peer review.

• The authors do not describe the difference between mir27b-5p and mir27b-3p. This needs to be better described.

We thank the reviewer for this comment and have included the following information in the revised manuscript.

“The precursor miR-27b has two mature arms: miR-27b-5p and miR-27b-3p that originates from the 5’ or 3’ strand of the precursor miR-27b, respectively and are almost complementary to each other. The stability and functionality of each arm depends on the tissue/cell type and according to miRBase Sequence database ^1^ (database of published miRNA sequences and annotation), miR-27b-3p (98.8%) is more abundant than miR-27b-5p (1.2%) and therefore is assumed to be the stable and functional version.”

• Why does mir27b-5p decrease NDUFS4 compared to control in Figure 2J?

As stated in response to #3, since *miR-27b-5p* and *miR-27b-3p* are almost complementary to each other, it is not surprising that *miR-27b-5p* decrease NDUFS4. However, as stated in response to #3, only *miR-27b-3p* is abundant in vivo.

• It's not clear why the authors used MitoCarta 2.0 and not the more updated MitoCarta 3. An explanation for this decision would be appreciated. Regardless, it would be ideal to retroactively expand the analysis to MitoCarta 3.0.

We have now retroactively expanded the analyses to MitoCarta 3.0. Overall, the reanalysis did not significantly deviate from our original findings. We identified a total of 848 nuclear-encoded mitochondrial proteins (original 840 proteins), but we did not uncover new *trans­­*-pQTL hotspots. Except the chr7 locus, the other two loci remained the same. Two proteins (MDH1 and TKT) disappeared from the chr7 locus, and one new protein (LDHB) appeared; however, the peak *trans­­*-pQTL SNP (rs32451909) and the candidate protein (COQ7) remained the same. These were reflected in revised Figures 4A-C. With respect to WGCNA, the same 5 modules reappeared with minor changes in protein composition. These were reflected in revised Figures 1B, 2B, 3B and 4B and in the main text.

Figure 2• The correlation between miR-27b expression and heart weight is not impressive. Given that this is a key part of the mechanistic hypothesis underlying this correlative observation and the low magnitude and lack of statistical power of the effect in Figure 2H, this is a significant weakness.

The statistics included a few strains not belonging to either ‘AA’ or ‘GG’ genotypes but to an ‘AG’ group. After removing these, the correlation and P values were improved (bicor = -0.267 and p value = 0.02).

• Apart from changes in heart size, it would be informative to assess hemodynamic cardiac characteristics such as ejection fraction to assess changes in cardiac function.

We appreciate the reviewer’s comments. However, this is a population-based study of over 72 strains of mice and it was not feasible to perform detailed hemodynamic cardiac characteristics. The goal of the study was to understand the genetic regulation of heart hypertrophy and we feel that our current findings are significant in several respects. First, we identified three novel *trans*-regulatory genetic loci that control distinct classes of mitochondrial proteins as well as heart hypertrophy using an integrative proteomics approach. Each of these loci contains mitochondrial proteins previously shown to affect heart pathophysiology but by unknown mechanisms (*miR-27b*/NDUFS4 ^2-4^, LRPPRC ^5^ and COQ7 ^6^). Also, each of these three *trans*-regulating hotspots were specific for the proteome, as they were not identified in our heart transcriptome. Further, we used both experimental and statistical analyses to support their causal roles. Our results provide evidence that common variations of certain mitochondrial proteins can act in *trans* to influence mitochondrial functions and contribute to heart hypertrophy. Of course, in-depth investigations such as hemodynamic cardiac assessments would be informative, but we humbly believe it is beyond the scope of the current paper.

• Changes in protein abundance don't necessarily indicate changes in function, especially such small changes. Therefore, the addition of functional assays directly related to the protein would greatly enhance this paper. For example, the authors could examine changes in mitochondrial respiration or metabolism between treatments and genotypes.

As in the previous point, we believe it is beyond the scope of the current paper. We emphasize that this is a population-based study of over 72 strains of mice, and it was not feasible to perform detailed cardiac characteristics.

• As a result of these issues, the authors' conclusion in lines 201-202 is not, in my opinion, adequately supported by the provided data. Figures 2H and 2I show a correlation. The cause-and-effect experiment of transfecting miR-27b into NRVM is not very impressive by itself. Adding a third replicate to the PE group may help show if there is a statistical difference; it's not clear why they have fewer replicates in this group. Primary ACM might be an alternative model system to show miR-27b increases NDUFS4 levels (although it would be more technically challenging). The researchers could also block miR-27b and if their hypothesis holds true should see a decrease in NDUF4.

As stated in response #2 listed for essential revisions, we have independently repeated our experiments and have now revised the data with new blots and graphs (Figure 2J). We have also supplied source data of the blots as part of the peer review.

Figure 3• The lack of any functional assessments to strengthen the correlation, let alone actual causation experiments is clearly apparent. Please make the data for this figure to provide a functional assessment.

This is primarily a genetic study, that was intended for the Systems Genetics issue of *eLife*, rather than a physiological study of engineered mouse models. The findings relate to the identification of regulatory loci and pathways controlling a phenotype relevant to heart function. Of course, analysis of aspects such as ejection fraction would be important in further understanding the underlying mechanisms, but we feel that this is beyond the scope of the paper.

OverallIn lines 44-46 the impression was given that these three variations would cause heart weight changes in both ISO and DIO HF models. Unfortunately, only one of the three genes showed heart weight changes in both conditions.

We have now revised the main text.

“Variations of all three were associated with heart mass in one of two independent heart stress models, namely, isoproterenol (ISO)-induced heart failure and diet-induced obesity (DIO) models.”

Tissue slices showing the histology or functional imaging studies of the heart (both of which have been done previously by papers published by this group) would provide much stronger evidence that the genes they identify could play a role in causing/preventing heart failure.

We appreciate the reviewer’s comments but emphasize that this is a population-based study of over 72 strains of mice, and it was not feasible to perform detailed cardiac characteristics. To go back and repeat the entire study, involving hundreds of strains and transcriptomics and proteomics analyses, is simply not feasible.

Are the changes in weight the authors see sufficient to cause/prevent heart failure? It would be a nice touch for the authors to include additional comments in this area in the discussion as some of the changes in weight are small.

We have now included percent changes between the two genotypes to better reflect the changes quantitatively.

• The size effect impact on cardiac hypertrophy is more difficult to appreciate as if I understand well only the data after ISO or DIO are shown for each allele but not baseline values. LV mass/body weight % of 0.4-0.45% as shown in Figure 2D does not suggest massive hypertrophy (0.4% = 100 mg LV for 25 g of body weight, for instance, can be seen in normal mice). The same holds true for figures 2E-F. I would like the authors to give the values of LV mass in untreated animals (is it similar between the 2 genotypes?) and maybe express the hypertrophy in relative increase to baseline in each group. In this way, one could better realize if the treatment was really efficient to trigger hypertrophy and how much the response differs between the groups.

We have now revised all figures to include both the control and ISO groups and the percent changes between the two genotypes in each group to better reflect the changes quantitatively.

• The same holds true for figures 3E-F and 4E-F: Is this clinically meaningful as the size effect seems to be very low? I would like the authors to discuss that point.

We have now included percent changes between the two genotypes to better reflect the changes quantitatively but since heart failure is undoubtedly a complex trait, involving hundreds of genes each with very small effect sizes, it is important to characterize modest as well as large effects. We now discuss this point in the manuscript.

“Nevertheless, the percent change in heart mass between the three loci were only of modest value. However, both heart failure and hypertrophy being complex traits are influenced by a large number of genetic factors, each exerting a small to large effect. The resulting phenotypes are therefore a sum of all impacts. So, we believe it is important to characterize modest as well as large effects.”

Can the 3 loci have a cumulative effect? Is the study powered enough to detect that?

Yes, the 3 loci have an additive effect. The analysis we did was the following:

a) For each of the three loci, the mean heart weight (both raw and normalized) for each of the genotypes were calculated, and then the difference between the highest and lowest homozygous groups were found.

b) Sum these differences for all three of the loci to get the expected difference between the genotype combination with the lowest theoretical heart weight and the one with the highest theoretical heart weight.

c) This expected difference to the heart weight from the genotype combination were then added with the lowest theoretical heart weight to see how different it is from the heart weight from the genotype combination with the highest theoretical heart weight.

For example, from the chow dataset, the differences in raw heart weight for the homozygous groups are the following:

chr7: 0.013 (0.127 [GG] – 0.114 [AA])

chr13: 0.015 (0.128 [AA] – 0.113 [GG])

chr17: 0.004 (0.120 [GG] – 0.116 [TT])

which leads to an expected shift of ~0.032.

The AA, GG, TT combination has a mean heart weight of 0.112, so the expected heart weight for the GG, AA, GG combination is 0.144. The actual weight is 0.139, so taking (observed – expected) / expected * 100 gives an error of ~5%.

For raw heart weights, the % errors are as follows:

chow_female HFHS_female HFHS_male Iso_treated

-5.017749 -2.902519 -7.117292 -3.281568

For normalized heart weights, the % errors are as follows:

chow_female HFHS_female HFHS_male Iso_treated

-6.993903 -2.239902 1.728702 -2.803209

This is consistent with an additive model.

• Could the authors provide the level of other markers of hypertrophy from their protein screen such as BNP, Β-MHC (although difficult to distinguish from α-MHC by proteomic), or any other relevant protein? This would strengthen their hypothesis if the loci variant were also associated with quantitative differences in these classical markers of cardiac hypertrophy.

We computed correlations between the three candidate proteins and known hypertrophic markers (NPPA and MYH7) and the results are included in table S10 in the manuscript.

• sum up the values of mitochondria and lysate levels per individual to perform a statistical analysis of the total level of each coQ?

We have revised Figure 2J to include the sum of mitochondria and lysate levels and they were still significant between the genotypes.

• If we admit that the total CoQ9 and CoQ10 levels are increased when COQ7 involved in their synthesis is reduced, it remains a puzzling observation. The authors discuss the possibility that CoQ is relocalized to the outer membrane or in the cytosol when COQ7 is reduced in heterozygotes CO7 mutant or in the mice bearing the rs32451909 GG allele, but this does not resolve the question of the total level of CoQ metabolite. Can the authors elaborate a bit further on that observation and by which mechanism it could occur? For instance, are the proteins involved in coQ degradation repressed when COQ7 is low?

We agree with the reviewer that this is puzzling but do not believe that these details are known.

Reviewer #2 (Recommendations for the authors):Figure 3– The authors could comment on whether patients with Leigh syndrome, French-Canadian type suffer from cardiomyopathies.

LSFC patients usually present symptoms during the first year of their life and their average life expectancy is four to six years. However, in some cases, the patients exhibit cardiomyopathies. We have now included the following in the main text as a response to the reviewer’s comment.

“Loss of function mutations in LRPPRC cause a congenital mitochondrial disease called Leigh syndrome, French-Canadian type that is often characterized by mitochondrial complex IV deficiency and impaired mitochondrial respiration and in some cases, neonatal cardiomyopathy and congenital cardiac abnormalities have been reported ^5,7^.”